# Efficient Core-set Selection for Deep Learning Through Squared Loss Minimization

Jianting Chen [1]

## Abstract

Core-set selection (CS) for deep learning has become crucial for enhancing training efficiency and understanding datasets by identifying the most informative subsets. However, most existing methods rely on heuristics or complex optimization, struggling to balance efficiency and effectiveness. To address this, we propose a novel CS objective that adaptively balances losses between core-set and non-core-set samples by minimizing the sum of squared loss across all samples. Building on this objective, we introduce the Maximum Reduction as Maximum Contribution criterion (MRMC), which identifies samples with the maximal reduction in loss as those making the maximal contribution to overall convergence. Additionally, a balance constraint is incorporated to ensure an even distribution of contributions from the core-set. Experimental results demonstrate that MRMC improves training efficiency significantly while preserving model performance with minimal computational cost.

## 1. Introduction

Data-driven deep learning heavily depends on large-scale datasets, but the growing size of models has led to exponentially increasing computational costs. This issue is particularly critical in resource-constrained scenarios, making efficient learning an area of rising interest (Shen et al., 2023). Core-set selection (CS), one of the tasks in efficient learning, focuses on selecting a small, yet representative subset (core-set) from the original dataset to reduce sample size while preserving model performance (Katharopoulos & Fleuret, 2018; Coleman et al., 2020). By optimizing data utilization, CS plays a crucial role in alleviating the tension between rapidly growing data scales and constrained resources.

CS is often regarded as a static data pruning technique, in contrast to dynamic data pruning (DDP) (Raju et al., 2021; Nguyen et al., 2023; Qin et al., 2024). Both aim to enhance training efficiency but differ significantly: CS retains only selected data, while DDP requires access to the full dataset. Moreover, CS focuses on data value, whereas DDP emphasizes the model and training process. While both are valuable, this work mainly focuses on CS.

Mainstream CS methods can be categorized into four types: ① Geometry-based methods analyze data distribution in the feature space and select representative samples by clustering algorithms, that cover the data distribution, ensuring diversity and consistency (Sorscher et al., 2022; Xia et al., 2023). ② Uncertainty-based methods prioritize samples that provide the highest information gain based on the model's prediction uncertainty. Heuristic metrics include forgetting events (Toneva et al., 2019), gradient norms (Paul et al., 2021), loss variance (He et al., 2024), and decision boundary errors (Yang et al., 2024). ③ Optimization-based methods frame CS as a mathematical optimization problem, aiming to maximize the core-set's ability to substitute the original dataset from the perspective of model parameter optimization (Killamsetty et al., 2021b;a). ④ Hybrid methods combine geometric properties and uncertainty metrics to balance diversity and information gain (Zheng et al., 2023).

With the rise of large-scale models like large language models and multimodal systems, CS has gained new importance. Recent studies focus on tailoring CS techniques to the specific characteristics of data and models, optimizing both training and fine-tuning processes (Xia et al., 2024a; Wang et al., 2024b; Evans et al., 2024).

Despite significant progress in CS, several challenges remain to be addressed. First, balancing diversity and uncertainty remains challenging. Diverse samples reflect the global distribution, while uncertain samples emphasize local information gain. Diversity can be understood as the uncertainty of the overall distribution given known local information. Therefore, both factors are essential for a core-set, but their conflicting objectives complicate achieving an effective balance. Second, there is a gap between heuristic and principled methods. Heuristic methods are simple but

---

[1]Department of Computer Science and Technology, Tongji University, Shanghai, China. Correspondence to: Jianting Chen <tj_chenjt@tongji.edu.cn>.

*Proceedings of the 42nd International Conference on Machine Learning*, Vancouver, Canada. PMLR 267, 2025. Copyright 2025 by the author(s).

lack theoretical rigor, while principled methods are resource-intensive and often fall short in practical applications.

To address these challenges, we propose a novel CS objective that replaces complex bi-level optimization strategies in principled methods with a simpler goal: to minimize the sum of squared loss of all samples, i.e., $\min \Sigma (l_i)^2$. The L2 norm amplifies the penalty on high-loss samples, preventing overfitting to the core-set and promoting generalization to the non-core-set, achieving an adaptive balance between the two. Assuming consistent initial loss values, the objective can be reformulated into two components: maximizing loss reduction and balancing loss reduction.

Building on this objective, we introduce the Maximum Reduction as Maximum Contribution criterion (MRMC), which leverages the concept of loss reduction attribution to evaluate mutual sample contributions to loss reduction. MRMC addresses the two components as follows: ① Maximizing loss reduction prioritizes samples with significant contributions, identified by their loss reduction during the initial training process. Samples with high contribution are those that substantially reduce the overall dataset loss. ② Balancing loss reduction employs a proxy-based regularization to adjust selection weights, ensuring balanced contributions and promoting diversity. The MRMC criterion is simple to implement and is theoretically inspired.

We evaluate MRMC on image recognition tasks, demonstrating strong performance in low-resource settings and consistently outperforming other methods. Parameter sensitivity analysis further confirms the effectiveness of the criterion and regularization technique.

The contributions of our work are as follows:

1. We propose a novel CS objective based on minimizing the sum of squared loss, which balances convergence between the core-set and non-core-set samples while being intuitive and easy to solve.

2. We introduce the concept of loss reduction attribution to quantify sample interactions, bridging the gap between the objective and the solution algorithm.

3. We propose the MRMC criterion to identify high-contribution samples and introduce a regularization to ensure balanced contributions. Both approaches are easy to implement.

4. Experiments on image recognition tasks demonstrate its superior performance across various core-set sizes and validate the reliability of its key modules and parameters.

## 2. Related Work

**Geometry-based methods** leverage the spatial structure of sample features to select core-sets that preserve data diversity and distribution consistency. Theoretically, diversity can enhance model robustness by increasing intra-class feature variance (Yu et al., 2020; Chan et al., 2022). Representative methods include clustering algorithms based on Euclidean distance, such as K-center (Sener & Savarese, 2018) and K-means (Sorscher et al., 2022), which improve feature diversity by removing redundant samples. Additionally, Moderate-DS (Xia et al., 2023) prioritizes samples near the median of intra-class centers to maintain distribution consistency. FastCore (Chai et al., 2023) ensures that the core-set retains gradient information by maximizing the distances between a sample and others within the same cluster. However, these methods heavily rely on the reliability of feature representations. Low-quality or insufficiently trained encoders are likely to undermine their effectiveness.

**Uncertainty-based methods** assume that samples with higher uncertainty contain more information and thus have greater learning value. Classical uncertainty measures, such as Least Confidence (Wang & Shang, 2014) and Entropy (Coleman et al., 2020), are widely utilized in various applications. In CS methods, forgetting events (Toneva et al., 2019) capture samples that are repeatedly forgotten by the model. EL2N and GraDN (Paul et al., 2021) quantify the uncertainty of the sample through prediction errors and gradient norms. Additionally, the distance between a sample and the decision boundary is another critical measure of uncertainty (Liu et al., 2019). (Yang et al., 2024) proposes selecting samples near the decision boundary to preserve the original decision boundary. Although uncertainty-based methods are simple and efficient, their heuristic nature often lacks a rigorous theoretical justification. Furthermore, noisy samples may also exhibit high uncertainty, potentially interfering with the performance.

**Optimization-based methods** focuses on model optimization, aiming to ensure that the contribution of the core-set to parameters is consistent with that of the original dataset. These methods typically rely on sample gradient information. CRAIG (Mirzasoleiman et al., 2020) and GradMatch (Killamsetty et al., 2021a) select samples by minimizing the gradient difference between the core-set and the original dataset. ADACORE (Pooladzandi et al., 2022) employs exponentially weighted Hessian approximations to estimate loss curvature and construct the core-set. RCS (Xia et al., 2024b) employs lexicographic optimization to balance model performance while reducing data size. Another widely used tool is Data Shapley (Ghorbani & Zou, 2019), which quantifies each sample's contribution to the model and selects the core-set accordingly (Wang et al., 2024a). Although optimization-based methods are theoret-

ically well-founded, their high computational complexity poses significant challenges in practice.

**Hybrid methods** aim to integrate the strengths of various approaches. For example, (Sorscher et al., 2022) incorporates geometric distances and uncertainty metrics to improve the selection process. D2Pruning (Maharana et al., 2024) employs a graph-based message-passing mechanism, where sample scores are iteratively refined based on the difficulty scores and feature distances of neighboring nodes. MolPeg (Chen et al., 2024) adopts a dual-model training strategy to select both representative and challenging samples. By combining diverse characteristics, hybrid methods not only enhance the core-set quality but also offer a fresh perspective for developing unified theories and frameworks.

## 3. Preliminary

Consider a dataset $D = \{z_i\}_{i=1}^n = \{(x_i, y_i)\}_{i=1}^n$, consisting of $n$ samples, where each sample $z_i$ is composed of an input $x_i$ and its corresponding label $y_i$. We aim to train a deep learning model parameterized by $\theta$ by minimizing a loss function $\mathcal{L}$. The training process employs mini-batch stochastic gradient descent (SGD) to update the parameters over $K$ steps. Let $\theta^t$ denote the parameter values after the $t$-th update, and let the loss for sample $z_i$ at these parameter values be $l_i^t = \mathcal{L}(z_i, \theta^t)$.

The task of CS, which is the focus of this study, involves selecting a subset $C \subset D$ from the dataset according to a selection ratio $\omega < 1$, such that $|C| = \lfloor \omega n \rfloor$. This subset $C$ serves as the core-set, replacing the original dataset for model training. The objective is to reduce the data size and improve training efficiency, while preserving model quality as much as possible.

## 4. Proposed Method

This section begins with the motivation for minimizing the sum of the squared loss of all samples, followed by the introduction of the concept of loss reduction attribution. Finally, it presents the MRMC criterion and the proxy-based regularization balancing technique.

### 4.1. Motivation and Objective

The training objective of deep learning models is typically to fit a given dataset $D$ by minimizing a loss function, which can be formalized as: $\min \sum_{z_i \in D} l_i^K$, where $l_i^K$ denotes the loss value of the $i$-th sample after $K$ iterations. In CS, only the core-set $C \subset D$ participates in training. However, the model must also generalize to the non-core-set samples $z_i \in D \setminus C$). Consequently, the objective is often defined

as a bi-level optimization problem:

$$\min_C \sum_{z_i \in D} \mathcal{L}(z_i, \theta_C^K),$$
$$\theta_C^K = \arg\min_{\theta^*} \sum_{z_i \in C} \mathcal{L}(z_i, \theta^*), \tag{1}$$

where $\theta_C^K$ is obtained by solving the inner optimization problem. Although bi-level optimization captures the objective of CS, it is computationally expensive because each outer-loop optimization requires solving the inner problem, which may be high-dimensional and non-convex. This contradicts the motivation of CS, which is to improve efficiency.

To address these challenges, we propose a simple and efficient first-order optimization objective. Specifically, the objective is reformulated as minimizing the squared loss over the entire dataset:

$$\min \sum_{z_i \in D} (\mathcal{L}(z_i, \theta_C^K))^2 = \min \sum_{z_i \in D} (l_i^K)^2. \tag{2}$$

The squared loss reduces overall loss while suppressing extreme values, balancing losses across samples, and improving generalization.

The final loss $l_i^K$ can be expressed as the initial loss minus the reduction in loss during training: $l_i^K = l_i^0 - \Delta l_i$. Assuming that the initial loss is constant across all samples ( $\forall z_i \in D, l_i^0 = l^0$), the objective can be rewritten as:

$$\min \sum_{z_i \in D} (l_i^K)^2 = \min \sum_{z_i \in D} (l^0 - \Delta l_i)^2. \tag{3}$$

Expanding and ignoring constants gives:

$$\min - \sum_{z_i \in D} \Delta l_i + \frac{1}{2l^0} \sum_{z_i \in D} (\Delta l_i)^2. \tag{4}$$

This objective function consists of two components:

- **The optimization term**: $-\sum_{z_i \in D} \Delta l_i$, which maximizes the total reduction in loss. This aligns with the fundamental goal of training deep learning models.

- **The regularization term**: $\frac{1}{2l^0} \sum_{z_i \in D} (\Delta l_i)^2$, which controls the balance of loss reduction. This prevents overfitting to the core-set samples, serving a role similar to regularization.

Notably, a similar optimization problem is also discussed in (Jain et al., 2023).

### 4.2. Loss Reduction Attribution

The objective function in Equation (4) uses the loss reduction $\Delta l_i$ as the variables. The reduction in loss is achieved

through updates to the model parameters $\theta$ using SGD. During the $t$-th parameter update, given a mini-batch $B^t$ and learning rate $\eta$, the parameter update is computed as:

$$\theta^t = \theta^{t-1} - \eta \sum_{z_j \in B^t} \nabla_\theta \mathcal{L}(z_j, \theta^{t-1}), \qquad (5)$$

where $\nabla_\theta \mathcal{L}(z_j, \theta^{t-1})$ represents the gradient of the loss for sample $z_j$ under the parameter $\theta^{t-1}$.

Based on first-order Taylor expansion approximations and the assumption of local smoothness, the change in the loss value for sample $z_i$ between consecutive updates, denoted as $\Delta l_i^t$, can be approximated as:

$$\begin{aligned} \Delta l_i^t &= \mathcal{L}(z_i, \theta^{t-1}) - \mathcal{L}(z_i, \theta^t) \\ &\approx \nabla_\theta \mathcal{L}(z_i, \theta^{t-1})^T (\theta^{t-1} - \theta^t) \\ &= \eta \sum_{z_j \in B^t} \nabla_\theta \mathcal{L}(z_i, \theta^{t-1})^T \nabla_\theta \mathcal{L}(z_j, \theta^{t-1}). \end{aligned} \qquad (6)$$

For simplicity, let us denote the gradient inner product as $G(z_i, z_j, \theta^{t-1}) = \nabla_\theta \mathcal{L}(z_i, \theta^{t-1})^T \nabla_\theta \mathcal{L}(z_j, \theta^{t-1})$.

On a larger scale, when the model is trained on the dataset $D$ over $K$ iterations, the total loss reduction of sample $z_i$ can be expressed as:

$$\begin{aligned} \Delta l_i &= \sum_{t=1}^{K} \Delta l_i^t = \eta \sum_{t=1}^{K} \sum_{z_j \in B^t} G(z_i, z_j, \theta^{t-1}) \\ &= \eta \sum_{z_j \in D} \left( \sum_{t=1}^{K} G(z_i, z_j, \theta^{t-1}) \mathbb{I}(z_j \in B^t) \right), \end{aligned} \qquad (7)$$

where $\mathbb{I}(z_j \in B^t)$ is an indicator function that equals 1 if $z_j$ is included in mini-batch $B^t$, and 0 otherwise. From the above equations, it can be observed that the loss reduction $\Delta l_i$ of sample $z_i$ is determined by the cumulative contributions of the gradient inner product $G(z_i, z_j, \theta^{t-1})$ from all training samples. This observation inspires the definition of a concept: **Loss Reduction Attribution**.

**Definition 4.1** (Loss Reduction Attribution). Assume a loss function $\mathcal{L}(\cdot, \theta)$, optimized over dataset $D$ through $K$ SGD updates. The loss reduction attribution expresses the total loss reduction $\Delta l_i$ of the sample $z_i \in D$ as the sum of attribution values (contributions) of all training samples $z_j \in D$, plus a negligible residual $\epsilon$:

$$\Delta l_i = \sum_{z_j \in D} a_{j,i} + \epsilon, \qquad (8)$$

where the attribution value $a_{j,i}$ is defined as:

$$a_{j,i} = \eta \sum_{t=1}^{K} G(z_i, z_j, \theta^{(t-1)}) \mathbb{I}(z_j \in B^t). \qquad (9)$$

Here, $a_{j,i}$ represents the contribution of the training sample $z_j$ to the loss reduction of $z_i$, thereby capturing the sample relationships during the learning process.

When training a model with the core-set $C$, the loss reduction attribution can expressed as $\Delta l_i = \sum_{z_j \in C} a_{j,i}$. Assuming the attribution values $\{a_{j,i}\}$ are stable (i.e., they do not vary significantly across different core-sets), and incorporating the objective function from Equation (4), the optimal core-set $C^*$ is defined as:

$$C^* = \underset{C \subset S, |C| = \lfloor \omega n \rfloor}{\arg\min} [\phi_{\text{opt}}(C) + \gamma \phi_{\text{reg}}(C)], \qquad (10)$$

where the optimization term $\phi_{\text{opt}}(C)$ and the regularization term $\phi_{\text{reg}}(C)$ are given by:

$$\phi_{\text{opt}}(C) = - \sum_{z_i \in D} \sum_{z_j \in C} a_{j,i}, \qquad (11)$$

$$\phi_{\text{reg}}(C) = \sum_{z_i \in D} \left( \sum_{z_j \in C} a_{j,i} \right)^2. \qquad (12)$$

The hyperparameter $\gamma$ controls the trade-off between the two terms. Specifically, the optimization term $\phi_{\text{opt}}(C)$ focuses on selecting samples that contribute most to the overall dataset's loss reduction. The regularization term $\phi_{\text{reg}}(C)$ ensures that contributions are distributed more evenly across all samples in the dataset, rather than being dominated by a few samples. To meet these two requirements, we will design algorithms to address them separately.

### 4.3. MRMC Criterion

To solve the optimization term $\phi_{\text{opt}}(C)$, we leverage the approximate symmetry of the attribution values. For any pair of samples $(z_i, z_j)$, we assume $|a_{j,i} - a_{i,j}| < \lambda_{i,j}$, where $\lambda_{i,j}$ denotes a permissible symmetry error. Referring to Equation (9), the gradient inner product used to compute the attribution values satisfies the commutative property, i.e., $G(z_i, z_j, \theta^{(t-1)}) = G(z_j, z_i, \theta^{(t-1)})$. However, the symmetry may be affected by the indicator function $\mathbb{I}(z_j \in B^t)$. Specifically, if $z_i \in B_a$ and $z_j \in B_b$ during an epoch, their gradient inner products are calculated using different parameters $\theta^{a-1}$ and $\theta^{b-1}$. Such differences may cause deviations in the gradient inner products. Nevertheless, under small learning rates in SGD, parameter updates are minimal, allowing us to approximate attribution values as symmetric.

Based on this assumption of approximate symmetry, the optimization term can be reformulated as:

$$\begin{aligned} \phi_{\text{opt}}(C) &= - \sum_{z_i \in D} \sum_{z_j \in C} a_{j,i} = - \sum_{z_j \in C} \sum_{z_i \in D} a_{j,i} \\ &= - \sum_{z_j \in C} \sum_{z_i \in D} (a_{i,j} \pm \lambda_{i,j}), \end{aligned} \qquad (13)$$

and by ignoring the symmetry error (i.e., $\lambda_{i,j} = 0$), and referring to Equation (8), we derive:

$$\phi_{\text{opt}}(C) \approx - \sum_{z_j \in C} \Delta l_j. \qquad (14)$$

This linear additive relationship suggests that minimizing $\phi_{\text{opt}}(C)$ can be achieved by selecting samples with the largest $\Delta l_j$, as these samples tend to contribute most significantly to other samples during training. We thus propose the **Maximum Reduction as Maximum Contribution** criterion (MRMC), expressed as: $\varphi_{\text{MRMC}}(z_j) = \Delta l_j$.

To compute $\Delta l_j$, we argue that only a small number of training epochs on all samples is sufficient. As training progresses, a sample's self-contribution continuously accumulates and becomes significantly larger than its contribution to other samples, i.e., $a_{j,j} \gg a_{i,j}, \forall i \neq j$. However, CS places greater emphasis on a sample's contribution to other samples. Therefore, only minimal training is required to measure sample contributions, aligning with the motivation of CS to improve efficiency.

Specifically, we train the model on the original dataset $D$ for $R$ epochs to compute $\varphi_{\text{MRMC}}(z_i)$, during which each sample $z_j$ is associated with a loss sequence $L_j = [l_j^{(1)}, l_j^{(2)}, \ldots, l_j^{(R)}]$. Due to the stochasticity of SGD, we avoid directly computing the difference between initial and final loss values. Instead, we adopt a simple mathematical model, assuming the loss values decrease following a negative exponential trend:

$$l_j^{(r)} \approx q_j \cdot w_j^{-r}. \qquad (15)$$

By fitting $q_j$ and $w_j$ to each loss sequence $L_j$, the MRMC criterion is computed as:

$$\varphi_{\text{MRMC}}(z_j) = \Delta l_j = q_j(1 - w_j^{-R}). \qquad (16)$$

Samples with higher MRMC criteria make greater contributions to the dataset and should be prioritized for selection.

### 4.4. Proxy-based Regularization

Minimizing the regularization term involves solving a quadratic optimization problem, which is challenging to address directly. To simplify, we further expand the regularization term as:

$$\phi_{\text{reg}}(C) = \sum_{z_i \in D} \left( \sum_{z_j \in C} a_{j,i} \right)^2 = |C| \sum_{z_j \in C} \sum_{z_i \in D} a_{j,i}^2$$
$$+ \sum_{z_j \in C} \sum_{z_k \in C} \sum_{z_i \in D} a_{j,i} a_{k,i} \mathbb{I}(k \neq j). \qquad (17)$$

The first term represents the $L_2$-norm of all attribution values within the core-set. The second term reflects the total pairwise interaction of attribution values between core-set samples across the entire dataset $D$. Our focus is on the second term, which captures the interrelations of attributions across samples.

Observing the attribution value based on gradient inner products in Equation (9), we find that a sufficient condition for the second term to zero is:

**Theorem 4.2.** *If the gradients of two samples are orthogonal, their attribution interaction is zero.*

*Proof.* The interaction between attribution values can be simplified by ignoring the indicator function term:

$$a_{j,i} a_{k,i} \propto G(z_i, z_j, \theta) G(z_i, z_k, \theta). \qquad (18)$$

Here, the gradient inner products $G(z_i, z_j, \theta)$ and $G(z_i, z_k, \theta)$ quantify the similarity of $z_j$ and $z_k$ to $z_i$ via their gradients. If the gradients of $z_j$ and $z_k$ satisfy orthogonality, then

$$\forall z_i \in D, G(z_i, z_j, \theta) G(z_i, z_k, \theta) = 0. \qquad (19)$$

The gradients of the core-set samples are expected to be orthogonal. This implies that the core-set samples must satisfy diversity constraints.

To achieve this, we propose a proxy-based regularization technique to enhance the diversity of the core-set. Specifically, we decompose the selection of core-set samples into a two-stage optimization process. In the first stage, we compute $\varphi_{\text{MRMC}}(z_j)$ as described in the previous section and select the top $\rho|C|$ ($\rho < 1$) samples with the highest scores to form an initial subset $C'$. Since these samples have high contributions, they are very likely to remain part of the core-set, even after considering the regularization term. In the second stage, we aim to enhance diversity by optimizing the selection of the remaining $(1 - \rho)|C|$ samples such that their gradient directions differ from those of the initial subset $C'$.

We train a small proxy model (using only the output layer) on the features $F_{C'}$ and labels $Y_{C'}$ of samples in $C'$. The model $\theta'$ is trained for a few epochs until convergence. For any remaining sample $z_i \in D \backslash C'$, the loss value is $\mathcal{L}(z_i, \theta')$. If the proxy model poorly fits $z_i$, it suggests that $z_i$ has gradient differences from the initial subset $C'$. Thus, the regularization score for the remaining samples is defined as:

$$\varphi_{\text{reg}}(z_i) = \exp\left( - \mathcal{L}(z_i, \theta') \right). \qquad (20)$$

The exponential operation normalizes the scores, thereby enhancing regularization performance in practice. By combining the MRMC criterion with the regularization constraint, we define the final scoring function as:

$$\varphi(z_j) = \varphi_{\text{MRMC}}(z_j) - \gamma \cdot \varphi_{\text{reg}}(z_j), \qquad (21)$$

where the $(1 - \rho)|C|$ samples with the highest scores are selected for inclusion in the core-set.

## 4.5. Selection Algorithm

Given a model and a loss function $\mathcal{L}(\cdot, \theta)$, as well as the entire dataset $D$, the detailed algorithm (see Algorithm 1) proceeds as follows:

**1. Initial training and observation:** First, the model is trained on the full dataset $D$ for $R$ epochs to observe the loss sequences across samples and obtain their feature representations. This step is standard for most CS algorithms. The advantage of our algorithm is that $R$ is very small, requiring minimal resources.

**2. MRMC criterion computation:** Using Equations (15) and (16), the MRMC criterion for each sample is calculated based on the observed loss sequence.

**3. CS without regularization:** If regularization is not applied, the top $|C|$ samples with the highest MRMC values are selected to form the core-set. The process ends here.

**4. CS with regularization:** If regularization is applied, an initial subset $C'$ is first constructed by selecting the top $\rho|C|$ samples with the highest MRMC values. A lightweight proxy model $\theta'$ is then trained on the samples in $C'$. For the remaining samples $D \setminus C'$, the top $(1 - \rho)|C|$ samples are selected based on their combined scores, computed using Equations (20) and (21). These selected samples are then added to the core-set.

**Efficiency Analysis.** Our method is characterized by high efficiency. It requires only a few epochs of model training, significantly reducing the cost of preparation. The selection algorithm operates with near $O(N)$ complexity and relies solely on each sample's loss change, avoiding expensive pairwise distance calculations. Additionally, the proxy model for computing is a simple linear layer, which ensures extremely high training efficiency.

## 5. Experiments

This section includes four groups of experiments: ① Comparing MRMC with strong baselines; ② Verifying the effectiveness of balancing the losses; ③ validating the assumptions behind loss reduction attribution; ④ Sensitivity analysis of key parameters.

### 5.1. Experimental Setup

**Datasets.** We use four widely adopted datasets to evaluate the effectiveness of the proposed method, namely CIFAR-10, CIFAR-100, Tiny-ImageNet, and ImageNet-1k. The scale and complexity of these datasets increase progressively, with the number of classes ranging from 10 to 1000. These datasets enables the evaluation of CS algorithms under different task complexities.

**Models and training.** For CIFAR-10 and CIFAR-100,

---

**Algorithm 1** Core-set Selection Algorithm

---

  **Input:** Model $L(\cdot, \theta)$, dataset $D$, core-set size $|C|$, hyper-parameters $\{R, \rho, \gamma\}$
  **Output:** Core-set $C$
  **for** $r = 1$ **to** $R$ **do**
    Update $\theta$ with mini-batch SGD on $D$
    Collect the loss values $\{l_i^{(r)} \mid z_i \in D\}$
  **end for**
  Compute $\{\varphi_{\text{MRMC}}(z_i) \mid z_i \in D\}$ using Eqs. (15)–(16)
  **if** $\rho = 1$ **then**
    $C \leftarrow \text{Topk}(D, |C|, \varphi_{\text{MRMC}}(z_i))$
  **else if** $\rho < 1$ **then**
    $C' \leftarrow \text{Topk}(D, \rho|C|, \varphi_{\text{MRMC}}(z_i))$
    Train a lightweight proxy model $\theta'$ using $C'$
    Compute $\{\varphi(z_i) \mid z_i \in D \setminus C'\}$ using Eqs. (20)–(21)
    $C \leftarrow C' \cup \text{Topk}(D \setminus C', (1 - \rho)|C|, \varphi(z_i))$
  **end if**

---

we use ResNet-18 (11.2M parameters) and train with a batch size of 128 for 200 epochs. For Tiny-ImageNet and ImageNet-1k, we use ResNet-34 (21.3M parameters) with a batch size of 256, training for 100 and 60 epochs, respectively. The optimizer is SGD across all datasets, with an initial learning rate of 0.1, momentum 0.9, and weight decay of $5 \times 10^{-4}$ (CIFAR) or $1 \times 10^{-4}$ (others). A cosine annealing schedule reduces the learning rate to a minimum of $1 \times 10^{-4}$, and data augmentation includes random horizontal flipping and cropping. The number of training epochs is fixed for all core-set sizes to ensure smaller core-sets reduce training time, aligning with the goal of efficiency.

**Competitors.** We compare the proposed method with the following classic or recently proposed CS algorithms[1]: **Random** selection; **EL2N & GraNd** (Paul et al., 2021) measures and selects samples with high uncertainty; **Glister** (Killamsetty et al., 2021b) formulates a bilevel optimization problem to maximize the log-likelihood on the entire dataset. **CCS** (Zheng et al., 2023) improves the coverage of core-set data by applying stratified sampling based on EL2N scores; **Moderate** prioritizes samples that are at intermediate distances from class centers; **Dyn-Unc** (He et al., 2024) selects samples with high uncertainty based on the standard deviation of loss values during the training process; **BoundaryCCS** (Yang et al., 2024) identifies the distance of samples to the decision boundary using adversarial learning and combines CCS to improve sample coverage; **D2pruning** (Maharana et al., 2024) computes sample difficulty based on EL2N scores and improves core-set diversity using a graph-based propagation method.

**Evaluation protocols.** We evaluate both the quality and efficiency of CS algorithm. **Core-set quality** is assessed

---

[1]Label balance constraint was not enforced in reproduction.

*Table 1.* Comparison of model accuracy (%) under two protocols for different core-set selection algorithms.

| Dataset | | CIFAR-10 (95.44) | | | | CIFAR-100 (79.37) | | | | Tiny-ImageNet (62.40) | | | |
|---|---|---|---|---|---|---|---|---|---|---|---|---|---|
| Selection ratio $\omega$ | | 70% | 50% | 30% | Avg | 70% | 50% | 30% | Avg | 70% | 50% | 30% | Avg |
| Full training protocol | Random | 94.45 | 93.60 | 91.06 | 93.04 | 76.09 | 72.83 | 67.01 | 71.98 | 58.14 | 52.43 | 45.25 | 51.94 |
| | Glister | 94.80 | 94.20 | 90.15 | 93.05 | 76.80 | 73.21 | 66.90 | 72.30 | 57.20 | 55.05 | 48.23 | 53.49 |
| | Moderate | 94.25 | 92.43 | 89.22 | 91.97 | 76.38 | 73.22 | 67.14 | 72.25 | 57.54 | 52.94 | 44.32 | 51.60 |
| | Dyn-Unc | 91.50 | 89.59 | 85.82 | 88.97 | 73.91 | 70.09 | 64.06 | 69.35 | 52.98 | 44.43 | 35.57 | 44.33 |
| | Boundary | 94.72 | 93.18 | 91.21 | 93.04 | 75.95 | 73.22 | 65.48 | 71.55 | 57.88 | 53.12 | 44.65 | 51.88 |
| | D2pruning | 92.76 | 91.31 | 88.97 | 91.01 | **77.79** | 74.10 | **69.65** | **73.85** | 60.05 | **57.37** | 49.04 | 55.49 |
| Early training protocol $R = 20$ | EL2N | 95.40 | 95.23 | 91.94 | 94.19 | 77.61 | 67.60 | 32.55 | 59.25 | 56.08 | 38.96 | 13.79 | 36.28 |
| | GraNd | 95.01 | 94.59 | 88.91 | 92.83 | 77.02 | 65.58 | 49.23 | 63.94 | 55.98 | 39.67 | 20.41 | 38.68 |
| | Glister | 94.05 | 92.59 | 89.10 | 91.91 | 76.39 | 72.49 | 66.05 | 71.64 | 56.50 | 53.98 | 47.68 | 52.72 |
| | CCS | 95.40 | 95.03 | 92.65 | 94.36 | 75.31 | 71.15 | 64.63 | 70.36 | 56.51 | 51.46 | 43.72 | 50.56 |
| | Moderate | 94.20 | 92.94 | 91.51 | 92.88 | 76.27 | 72.97 | 66.58 | 71.94 | 57.32 | 52.01 | 44.37 | 51.23 |
| | Dyn-Unc | 92.72 | 87.86 | 79.47 | 86.68 | 75.29 | 67.05 | 47.78 | 63.37 | 56.66 | 39.97 | 12.57 | 36.40 |
| | Boundary | 94.50 | 93.68 | 92.08 | 93.42 | 75.94 | 72.16 | 66.10 | 71.40 | 57.76 | 53.01 | 45.01 | 51.93 |
| | D2pruning | 92.22 | 90.89 | 87.95 | 90.35 | 75.97 | 73.56 | 69.21 | 72.91 | 59.42 | 56.88 | 50.15 | 55.48 |
| | MRMC | 94.94 | 94.61 | 92.27 | 93.94 | 77.76 | 74.43 | 67.28 | 73.16 | 59.89 | 56.89 | 49.99 | 55.59 |
| | MRMC-R | **95.46** | **95.24** | **93.13** | **94.61** | 77.17 | **74.46** | 68.66 | 73.43 | **60.42** | 57.03 | **50.88** | **56.11** |

The best results are highlighted in **bold**, and the second-best results are indicated with an underline.

by the accuracy of the model on the test set after training on the selected core-set. Following standard practices, we prepare core-sets of varying sizes and compare the model's performance trained on core-sets selected by different algorithms. Higher test accuracy indicates better core-set quality. For **selection efficiency**, the main computational cost arises from training the model prior to selection to obtain sample uncertainties or feature representations. Methods such as MRMC and EL2N suggest this cost can be reduced by training for only a few epochs. Hence, we adopt two protocols: (1) **Full training protocol:** CS is performed after fully training the model (all epochs). (2) **Early training protocol:** CS is performed after training the model for only $R = 20$ epochs. Core-set quality is evaluated under both protocols for all algorithms except EL2N, CCS, and MRMC. Most experiments are conducted three times with different seeds.

**Hyperparameters.** For MRMC, the number of initial training epochs is set to $R = 20$. For the regularized version (MRMC-R), the regularization parameters are set to $\rho = 0.3$ and $\gamma = 2$. The proxy model is trained with SGD (learning rate 0.01, batch size 512) for 10 epochs. For competitor algorithms, we follow the publicly available hyperparameter settings during reproduction.

### 5.2. Comparison with Baselines

We compare MRMC with other CS algorithms across the CIFAR-10/100 and Tiny-imagenet datasets. Table 1 presents the model performance under different core-set sizes.

On CIFAR-10, MRMC-R achieves the best performance across all core-set sizes, with CCS as the second-best.

D2pruning performs differently from the original results (Maharana et al., 2024), likely due to differences in training iterations. Under the early training protocol, Moderate and Boundary outperform, indicating that full training model may lead to overfitting on simpler tasks.

*Table 2.* Comparison results on ImageNet-1K.

| Selection ratio | 50% | 30% | 10% | Avg |
|---|---|---|---|---|
| Random | 68.51 | 64.57 | 48.26 | 60.45 |
| D2pruning | 68.71 | 65.26 | **52.14** | 62.04 |
| MRMC | **69.81** | **66.47** | 50.48 | **62.25** |
| MRMC-R | 69.85 | 66.05 | 49.66 | 61.85 |

On CIFAR-100, D2pruning achieves the best results, followed by MRMC-R. However, under the early training protocol, most methods degrade, while MRMC-R becomes the best. Even the basic MRMC without regularziation also outperforms others. Notably, when compared to EL2N and Dyn-Unc, both of which are similarly simple algorithms that observe and leverage outputs from the model during training, our MRMC criterion proves to be more effective.

The results on Tiny-ImageNet are similar to those on CIFAR-100. Both MRMC and D2pruning demonstrate superior performance. This suggests that for more complex tasks, it is critical for CS to balance diversity and uncertainty. Furthermore, employing the MRMC criterion alone can achieve comparable outcomes, as it inherently adapts to balance diversity and uncertainty.

The results on ImageNet-1K under the early training protocol are shown in Table 2. With a fixed training of 60 epochs, MRMC outperforms Random and D2pruning at selection

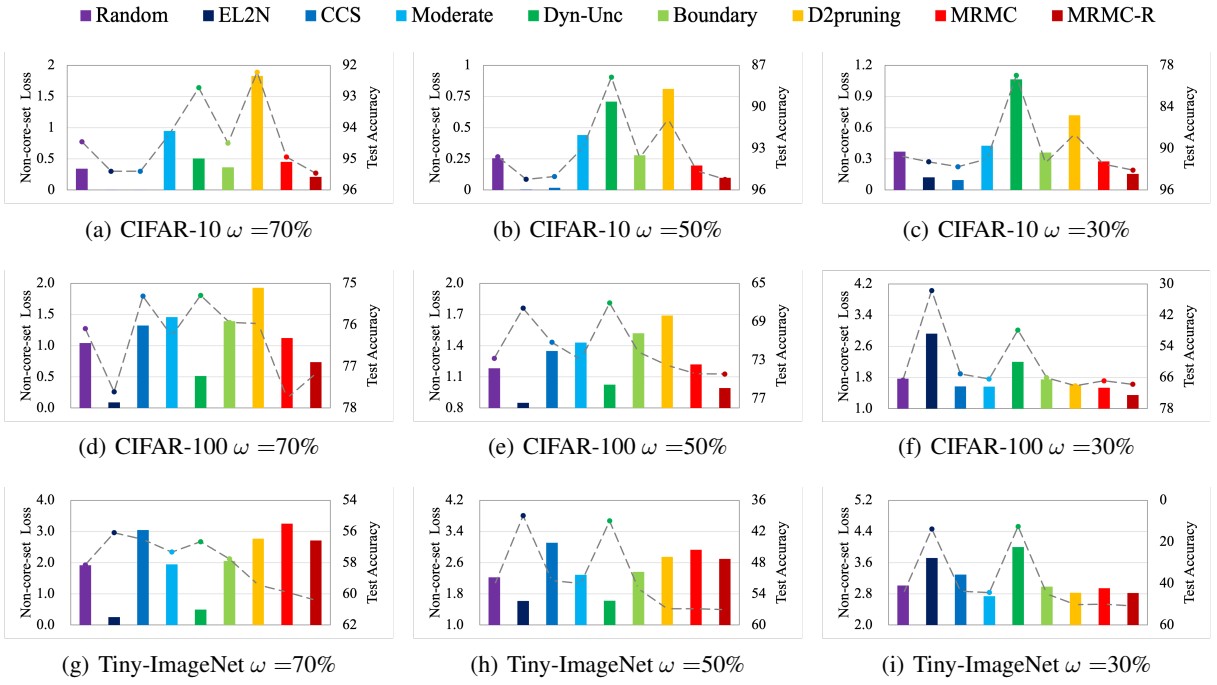

Figure 1. Relationships between non-core-set loss (lines) and test accuracy (bars) across different algorithms in the early training protocol.

ratios of 50% and 30%. The suboptimal performance of MRMC-R is likely due to hyperparameter settings.

For the time cost of sample selection on CIFAR-10/100 and Tiny-ImageNet, methods such as MRMC(-R), EL2N, GraNd, CCS, Moderate, and Dyn-Unc operate at a second-level scale with near-linear complexity. In contrast, Glister and D2Pruning require tens of seconds, while Boundary incurs a significantly higher cost, taking hundreds of seconds due to adversarial training. On ImageNet-1K, MRMC-R completes sample selection in approximately 1 minute, whereas D2Pruning takes around 6 minutes.

### 5.3. Effectiveness of Balancing Losses

The proposed MRMC and regularization techniques aim to reduce and balance the losses of core-set and non-core-set samples, allowing the model to better fit all data. Given that core-set losses are generally minimized, our focus shifts to the losses of non-core-set samples. Figure 1 shows the relationship between non-core-set loss and test accuracy.

For MRMC and MRMC-R (last two columns), regularization reduces non-core-set loss and improves model performance in most cases. With small core-sets ($\omega$ =30%), non-core-set loss directly correspond to model performance: lower loss yield better accuracy. However, on CIFAR-100 and Tiny-ImageNet with large core-sets, uncertainty-based methods (e.g., EL2N, Dyn-Unc) reduce non-core-set loss but degrade overall performance. In contrast, diversity-

enhanced methods (e.g., D2Pruning, MRMC) retain higher non-core-set loss while achieving superior accuracy. This highlights the importance of introducing diverse samples into the core-set for complex tasks.

### 5.4. Assumption Validation

We hypothesize that loss reduction attribution possesses stability and symmetry. To validate this, we conduct a case study. Specifically, we randomly sample 50 instances from CIFAR-100 and compute their attribution scores over 10 training epochs under four settings: using the full dataset and core-sets of 30%, 50%, and 70%. This results in four $50 \times 50$ attribution matrices, as shown in Figure 2.

For stability, we normalize the attribution scores and calculate the average relative error against the full-data attribution matrix. The relative errors for the 30%, 50%, and 70% core-sets are 27%, 26%, and 22%, respectively, indicating improved stability with larger core-sets.

For symmetry, visualizations of the attribution matrices consistently exhibit strong diagonal symmetry across all settings. The relative symmetry errors are 16%, 19%, 22%, and 24% for the 30%, 50%, 70% core-sets, and full dataset, respectively—showing that symmetry decreases as the amount of training samples increases.

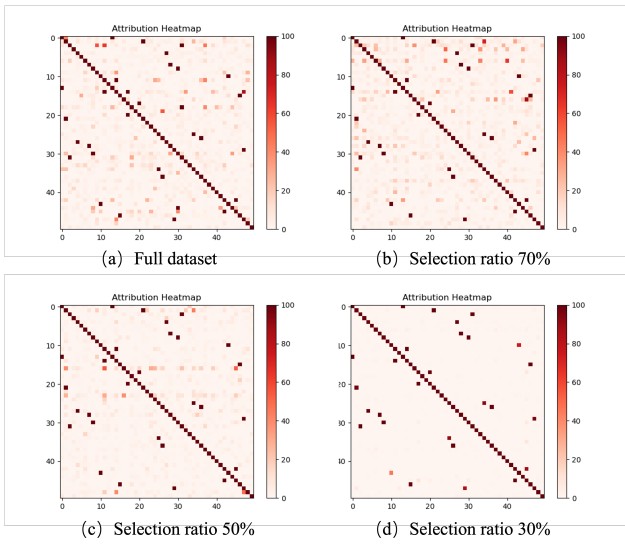

(a) Full dataset

(b) Selection ratio 70%

(c) Selection ratio 50%

(d) Selection ratio 30%

*Figure 2.* Visualization of loss reduction attribution values for the same 50 samples under training with core-sets of different sizes.

*Table 3.* Results of parameter sensitivity validation experiments.

| Dataset | CIFAR-100 | | Tiny-ImageNet | |
|---|---|---|---|---|
| Selection ratio | 70% | 30% | 70% | 30% |
| $R = 30$ | **77.83** | 66.38 | **60.25** | 49.66 |
| $R = 20$ | 77.76 | 67.28 | 59.89 | **49.99** |
| $R = 10$ | 76.52 | **67.82** | 58.74 | 48.73 |
| $R = 5$ | 75.51 | 66.33 | 57.67 | 46.76 |
| $\rho = 0.8$ | **77.81** | 66.98 | 58.87 | 47.2 |
| $\rho = 0.5$ | 77.21 | 68.26 | 58.94 | 49.83 |
| $\rho = 0.3$ | 77.17 | **68.62** | **60.42** | **50.88** |
| $\gamma = 5$ | **77.73** | 63.61 | **60.24** | 48.79 |
| $\gamma = 2$ | 77.21 | **68.26** | 58.94 | 49.83 |
| $\gamma = 1$ | 77.08 | 67.51 | 59.86 | **50.26** |

### 5.5. Parameter Sensitivity Analysis

We analyze the sensitivity of three key parameters on CIFAR-100 and Tiny-ImageNet datasets, using selection ratios of 70% and 30%. The parameters include the number of initial training epochs $R$, the scale of the regularization subset size $\rho$, and the trade-off parameter $\gamma$. The results are shown in Table 3.

**Effect of $R$.** Testing $R = 5, 10, 20$ and 30 with $\rho = 1$ (no regularization), most results show that larger $R$ improves core-set quality. However, for the 30% core-set on CIFAR-100, $R = 10$ performs best, while for the 30% core-set on Tiny-ImageNet, $R = 20$ performs best. This suggests that optimal $R$ depends on core-set size and task complexity.

**Effect of $\rho$.** For $\rho = 0.3, 0.5$, and 0.8 with $R = 20$ and $\gamma = 2$, smaller $\rho$ generally produces better performance, likely

due to better sensitivity to gradient differences. However, for the 70% core-set on CIFAR-100, $\rho = 0.8$ performs better, suggesting that larger initial subsets benefit from regularization to capture uncertain samples.

**Effect of $\gamma$.** We compare $\gamma = 1, 2$, and 5 with $R = 20$ and $\rho = 0.5$. The tradeoff parameter $\gamma$ balances the MRMC criterion and regularization. Results show that the optimal $\gamma$ varies with core-set size and task complexity.

## 6. Conclusion

We proposes a simple yet effective criterion (MRMC) for CS, which identifies samples contributing most to loss reduction, thereby enhancing the model to fit the known data. Compared to other algorithms, MRMC requires only limited initial training, offering advantages of simplicity and low computational cost. Additionally, we introduce a proxy-based regularization to enhance the diversity of core-sets. This approach adaptively balances the losses of core-set and non-core-set samples, improving the model performance.

**Limitations and future work.** We make several assumptions, such as the stability and symmetry of loss attribution values. Further optimization is possible if loss attribution can be efficiently computed and modeled. Furthermore, the regularization technique relies on multiple hyperparameters, which complicates practical implementation. Future work should focus on reducing this dependency for applicability.

## Acknowledgments

This work was completed during the visit to The Chinese University of Hong Kong. We sincerely thanks Professor Irwin King for his guidance. Gratitude is also extended to the co-advisor at Tongji University, Professor Yang Xiang, for his financial support.

## Impact Statement

This paper presents work whose goal is to advance the field of Machine Learning. There are many potential societal consequences of our work, none which we feel must be specifically highlighted here.

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
