# OpenReview forum: "Efficient Core-set Selection for Deep Learning Through Squared Loss Minimization"
_ICML.cc/2025/Conference — ICML 2025 poster_

### Official Review · Reviewer_opAA · 2025-02-14

**Overall Recommendation:** 3

**Summary:**

This paper proposes a two-phase core-set method for selecting a small but representative subset of training data. The first phase selects samples with the highest contributions, while the second phase employs a lightweight proxy model to evaluate the differences between the remaining samples and already selected samples, further selecting samples that can increase overall diversity.

# **update after rebuttal**
I would like to thank the authors for their sincere efforts to address my concerns. I have increase my score.

**Claims And Evidence:**

In my view, balancing diversity and uncertainty in core-set selection is indeed an important issue. However, the relative importance of diversity and uncertainty may vary in different scenarios. From my perspective, which admittedly may be incorrect, for moderate levels of pruning, more attention should on hard-to-learn samples. This ensures that all rare patterns, which can only be learned through memorization effects, are included in the training set. In this case, the optimization objective of the core set is to reduce redundant samples while retaining the information represented by all samples in the original dataset. On the other hand, in extreme pruning scenarios where very few samples are retained, the focus should be on easy-to-learn samples that represent simple and common patterns. This helps preserve the basic pattern information present in the original dataset in extremal setting.

Based on this perspective, maximizing loss reduction represents easy-to-learn samples, while balancing loss reduction attempts to incorporate hard-to-learn samples. Therefore, the authors' claims themselves are not problematic. However, the evidence provided by the authors is quite lacking:

1. The authors should provide very detailed ablation experiments for maximizing loss reduction and balancing loss reduction to evaluate their respective contributions. Instead, the authors devote a large portion of pages 3-6 to the formalized expression of the proposed method, which is neither critical nor important. The authors should use some of this valuable space (or include it in the appendix) to clearly experimentally verify the contributions of the two-stage method.

2. The authors should provide experiments with extreme removal rates similar to the baseline methods, such as the 90% pruning ratio used in D2pruning, to test the performance of their method in extreme removal settings (it's worth noting that this is precisely the setting where the authors' method has the most theoretical advantage).

3. The paper lacks a detailed evaluation of wall-time runtime. Even if the authors' method does not have an advantage in this aspect, they still need to conduct experiments to report the results.

**Essential References Not Discussed:**

I notice a minor oversight in the paper regarding the discussion and comparative experiments with Dynamic Data Pruning (DDP). Based on empirical evidence, DDP based methods demonstrate superior performance at moderate pruning rates, while static pruning methods regain their advantage under extreme pruning conditions. I strongly recommend that the authors incorporate discussions of the following papers and include the latest state-of-the-art DDP method in their baseline comparisons in Table 1:

[1] Accelerating deep learning with dynamic data pruning, arXiv 2021.

[2] KAKURENBO: Adaptively Hiding Samples in Deep Neural Network Training, NeurIPS 2023.

[3] InfoBatch: Lossless Training Speed Up by Unbiased Dynamic Data Pruning, ICLR 2024.

[4] Instance-dependent Early Stopping, ICLR 2025.

**Experimental Designs Or Analyses:**

See Claims And Evidence.

**Methods And Evaluation Criteria:**

See Claims And Evidence.

**Other Comments Or Suggestions:**

I suggest this paper currently falls slightly below ICML acceptance standards.
I should note that the 5-point scale used here (unlike the traditional 10-point system common in ML conferences) makes the evaluation confusion. I am assigning a Weak Reject, though my precise assessment would be a borderline reject. I hope the authors understand this current status more clearly.

In my view, the paper suffer from insufficient novelty in its contributions and inadequate experimental evaluation. I would direct the authors' attention to D2pruning, which shares similar motivations in addressing both DIVERSITY & DIFFICULTY, but provides substantially much more comprehensive experimental validation. To strengthen this work, the authors need to provide extensive additional experiments (I guess other reviewers will suggest specific evaluations as well) to demonstrate their method's contributions beyond D2pruning.

A minor suggestion: The term ''Selection ratio'' should be replaced with ''Pruning rate'' (30% Pruning rate = 70% Selection ratio) to maintain consistency with the established terminology in previous literature.

**Other Strengths And Weaknesses:**

This article is well-organized and the author attempts to address important challenge in core-set.

**Questions For Authors:**

Not applicable.

**Relation To Broader Scientific Literature:**

The authors proposed two-phases method position in two main approaches in **Dataset pruning**:

- **Static Selection**:
Performed before or in the early stages of training, aiming to identify a representative subset of the training data. Key research includes Core Set and Data Pruning studies by:

Huggins et al. (2016) Coresets for scalable...

Paul et al. (2021) Finding important examples early in training

Krishnateja et al. (2021) ``Glister''

Xia et al. (2022, 2024) on moderate and refined core set approaches

- **Dynamic Selection**:
Also known as Dynamic Data Pruning. Involves continuous sample selection throughout the training process. Notable works include:

Raju et al. (2021) Accelerating deep learning with dynamic data pruning

Truong et al. (2023) ``Kakurenbo''

Qin et al. (2024) InfoBatch

**Theoretical Claims:**

In the derivations presented on pages 3 to 6 of the paper, the authors primarily propose a conceptual framework to formalize their proposed method. I do not suggest  this constitutes theoretical contribution. While I think it is perfectly reasonable and unproblematic for the paper to lack theoretical contributions (definitely not affect my score), the weakness of the authors' experimental evidence leads me to expect some theoretical contributions from this paper. Moreover, as mentioned in the limitations section of the paper, the theoretical derivations in this work are based on too many assumptions.

---

> ### Author Rebuttal · Authors · 2025-03-30
>
> Your insights have greatly helped us identify key areas for improvement. We address the main concerns below:
> ### **Response #1: Insufficient Novelty in Contributions**
> Rather than introducing a new theoretical guarantee framework, our contribution focuses on the design of a simple, efficient, and practical selection strategy. MRMC can be computed with minimal overhead by measuring sample loss reduction. Although computing the OPT value requires training a proxy model, it is a single linear layer, making training extremely fast. The overall complexity is near-linear (≈O(N)).
>
> As the title suggests, “Efficient Core-set Selection” emphasizes not just coreset quality but also selection cost. E.g., MRMC uses only 20 epochs for preparation, whereas methods like D2Pruning require a fully trained (200-epoch) model on CIFAR. Moreover, D2Pruning involves costly feature distance computations and information propagation. On ImageNet-1K, D2Pruning takes 350 seconds to build a 30% core-set, while MRMC-R takes 50s and MRMC under 10s.
> ### **Response #2: Ablation Experiments**
> Table 1 already compares MRMC, which only maximizes loss reduction, with MRMC-R, which incorporates balanced loss regularization— the two core techniques in our framework. The regularized version shows clear advantages on more challenging tasks and under smaller coreset sizes. Section 5.3 specifically analyzes the effectiveness of the balancing term. An ablation study would be unnecessary.
> ### **Response #3: Extreme Pruning**
> We acknowledge this as a limitation of our work. As a heuristic method, we focus on selection efficiency, which comes at the cost of reduced performance under extreme pruning. While small subsets are highly valuable in  interpretability and incremental learning, our method is primarily optimized for moderate ratios, where it achieves comparable performance with low cost.
> ### **Response #4: Lack of Runtime**
> We appreciate your comment. Although efficiency is central to our work, we omitted a runtime analysis — this was an oversight. We will include wall-clock time  in the revision to better support our efficiency claims.
> As shown in the table, many strong SOTA methods suffer from high computational overhead, which limits their practicality on large-scale datasets. In contrast, MRMC and MRMC-R achieve a favorable balance between coreset quality and selection efficiency.
> Wall-clock time (seconds) for constructing a 50% coreset on CIFAR-10, CIFAR-100, and TinyImageNet, and a 30% coreset on ImageNet-1K:
> | Method     | CIFAR-10 | CIFAR-100 | TinyImageNet | ImageNet-1K |
> |-----|-----:|------:|--------:|-------:|
> | EL2N       |      0.1 |       0.1 |          0.2 |           — |
> | GraNd      |      0.1 |       0.1 |          0.2 |           — |
> | Glister    |     20.4 |      27.4 |         75.1 |           — |
> | CCS        |      0.3 |       0.3 |          0.9 |           — |
> | Moderate   |      0.8 |       0.8 |          1.5 |           — |
> | Dyn-Unc    |      0.2 |       0.3 |          0.7 |           — |
> | Boundary   |    102.1 |     127.1 |        290.7 |           — |
> | D2Pruning  |     10.9 |      11.3 |         25.1 |        352.9 |
> | MRMC       |      0.2 |       0.2 |          0.4 |          9.3 |
> | MRMC-R     |      1.0 |       1.3 |          2.9 |         50.7 |
> ### **Response #5:  Theoretical Assumptions**
> Our method is heuristic and not theoretically guaranteed. Nevertheless, we aim to ground our design in theoretically inspired approximations. Thus, we validate the assumptions empirically (see Response #2 to Reviewer JTrd). Our method aims to strike a balance between complex, inefficient theoretical methods and simple but less effective heuristics.
> ### **Response #6:  Comparison to DDP**
> We appreciate your suggestion and will include a discussion of Dynamic Data Pruning (DDP) methods in the Introduction. We agree that DDP methods (e.g., Kakurenbo, InfoBatch) have demonstrated strong performance in accelerating convergence.
> However, DDP and Static Data Pruning (SDP) differ fundamentally in both motivation and usage:
> - **SDP** selects a fixed subset  in the early phase of training and discards the remaining data. This enables savings not only in computation, but also in memory and storage, making SDP more suitable for on-device learning or communication bottlenecks.
> - **DDP** assumes access to the full dataset throughout training and aims to reduce computational cost via dynamic sample scheduling. DDP can be seen as an alternative to random data scheduling in SGD, and is more closely related to curriculum learning or advanced sampling strategies.
>
> Moreover, prior SDP works typically do not compare directly with DDPs, and we followed the same convention. Nonetheless, we agree that DDP is an important direction and will acknowledge it in the revision.
>
> ### **Summary**
> We hope that these responses address your concerns and help position MRMC as an effective  solution to the core-set selection problem.

---

> > ### Comment · Reviewer_opAA · 2025-04-02
> >
> > I would like to thank the authors for their sincere efforts to address my concerns. I am inclined to increase my score by +1 (as a result, 3). I would like to seeing all the revisions in the updated version.

---

### Official Review · Reviewer_uBL3 · 2025-02-19

**Overall Recommendation:** 3

**Summary:**

The authors:

*propose a new core-set selection approach that seeks to balance losses between chosen and unchosen samples by minimizing the overall sum of squared loss.

* Introduce the Maximum Reduction as Maximum Contribution (MRMC) criterion, which pinpoints those data points that most substantially reduce the loss—indicating they have the greatest influence on achieving model convergence.

* To maintain a fair representation, they impose a balance constraint so that the contributions of the core-set remain evenly distributed.


I like the idea of the paper and the writing style. My main concerns are related to the experiments and mainly competing methods,

**Claims And Evidence:**

1. The authors claim that prior work neglected the theoretical guarantees. However, theoretical methods for practical subset selection have been proposed already, both for data (https://proceedings.mlr.press/v202/tukan23a.html, https://ieeexplore.ieee.org/abstract/document/9941065) and model (https://proceedings.neurips.cc/paper_files/paper/2022/hash/f7fc38fdd95fd146a471791b93ff9f12-Abstract-Conference.html) compression.

**Essential References Not Discussed:**

x

**Experimental Designs Or Analyses:**

Experiments are not good enough as the authors does bt compete with SOTA baselines which achieve a much better accuracy. See cifar10/100 results in the paper "https://proceedings.mlr.press/v202/tukan23a.html," and all the other competing methods there.

**Methods And Evaluation Criteria:**

The authors evaluate using the accuracy metric on CIFAR10, 100, and Tiny ImageNet.
They also see how the loss on the corset is translated to accuracy in the test set.

They also analyze accuracy as a function of the three key parameters-> the number of initial training epochs, the scale of the regularization
subset size, and the trade-off between the losses parameter.

**Other Comments Or Suggestions:**

x

**Other Strengths And Weaknesses:**

Strengths:

1. good idea with some theoretical motivation -- the idea is that a good coreset should satisfy two things: diversity, and effect on the loss. So the authors compute the subset that contributes the most to reducing the loss, and add a regularization to constraint diversity of the coreset.

2. Writing is good.

Weaknesses:

Mainly experiments, prior work has some better results.

**Questions For Authors:**

You assume that “attribution values” (the pairwise gradient inner products) are stable and roughly symmetric. Have you measured how large the asymmetry (i.e., ∣aj,i−ai,j∣∣aj,i​−ai,j​∣) is in practice? Do you see any specific scenarios or datasets where this assumption might break?

**Relation To Broader Scientific Literature:**

x

**Theoretical Claims:**

1. Equation 1, why D\C and not the full data D?
2. I don't see how the second derivation works in (6), please provide the details carefully for review/

-----------------------
In general, the paper is not a theoretical one, while I agree it has a good theoretical *motivation*, it does not have theoretical guarantees. Thus I would not use the statements "balance efficiency and theoretical guarantees", "Both approaches are theoretically robust and easy to implement.", and many others -- I don't see theoretical guarantees on the training, approximation error, generalizability or else.

---

> ### Author Rebuttal · Authors · 2025-03-30
>
> Thank the reviewer for the feedback and positive comments on our idea and writing. Below we address the main concerns:
> ### **Response #1: Theoretical Guarantees in Prior Work**
> We realize that our original phrasing may have led to confusion. Our intention was to emphasize the practical gap between efficient heuristics and theoretically grounded methods, rather than to dismiss existing work with guarantees. We will update the manuscript to better reflect this nuance.
> ### **Response #2: Theoretical Claims**
> Following your suggestion, we reviewed the relevant literature and will modify Equation (1) to use D instead of D\C. As for Equation (6), we use a first-order Taylor expansion to approximate the change in loss, a standard approach in prior work. We avoid second-order terms to maintain algorithmic simplicity. We will add clarifications in the revised manuscript to justify this approximation.
> ### **Response #3: No Theoretical Guarantees**
> Thank you for pointing this out. We acknowledge  that our method does not provide formal guarantees on generalization or approximation error. Rather, it is a heuristic framework inspired by theoretical insights. We will revise the language to clearly state that our method is “theoretically inspired” rather than “theoretically guaranteed.”
> ### **Response #4: Comparison to SOTA Baselines**
> Beyond accuracy, a key contribution of our method is its selection efficiency. While achieving accuracy comparable to SOTA methods, our approach incurs significantly lower computational cost. We acknowledge the lack of runtime and complexity analysis in the original submission and will include it in the revision (see Responses #1 and #4 to Reviewer opAA).
>
> We also note that coreset evaluation protocols vary widely across prior work, making fair comparisons difficult. Key differences arise in two stages:
>
> - **Preparation before selection**: Some methods (e.g., *D2Pruning*, *Boundary-CCS*) require a fully trained model (e.g., 200 epochs) prior to selection, which is computationally expensive. In contrast, our method selects using a partially trained model (e.g., 20 epochs), offering substantial efficiency gains.
>
> - **Training after selection**:
>   - **Warm start**: Training proceeds without reinitializing the model after selection. This typically improves performance. We provide results showing our method further benefits from this tip.
>   - **Fixed iterations**: Keeping the number of SGD steps constant across core-set sizes nullifies the efficiency benefit of using smaller subsets.
>   - **Fixed epochs (ours)**: Using the same number of training epochs across different core-set sizes reduces the overall training cost as the size of the coreset decreases.
>
> Our experiments adopt the fixed-epoch, no-warm-start setting and include reproductions of the baseline methods for relatively fair comparison. Even under these constraints, our method—trained with only a few epochs—achieves comparable results to *D2Pruning*, which relies on full model training. Furthermore, our selection algorithm has near-linear time complexity (≈O(N)), making it scalable to large datasets such as ImageNet-1K.
>
> Regarding Tukan et al. (ICML 2023), as cited by the reviewer, their focus is on extremely small coreset sizes under a highly specific setup (learning rate 0.01, batch size 20, 300 epochs). This setting differs significantly from our experimental protocol and most existing data pruning benchmarks, making direct comparison less appropriate.
>
> | Method    | C10(70%) | C10(50%) | C10(50%) | C100(70%) | C100(50%) | C100(30%) | TI(70%) | TI(50%) | TI(30%) |
> |:----------------:|:--------:|:--------:|:--------:|:---------:|:---------:|:---------:|:-------:|:-------:|:-------:|
> | MRMC-R   | 95.46    | 95.24    | 93.13     | 77.17     | 74.46     | 68.66   | 60.42   | 57.03   | 50.88 |
> | MRMC-R-warm   | 95.65    | 95.25    | 93.88     | 77.83     | 75.00     | 70.05   | 61.44   | 58.46   | 55.01 |
>
>
> ### **Response #5: Empirical Validation of Key Assumptions**
> We appreciate the suggestion. We conducted a case study to empirically validate these assumptions (see Response #2 to Reviewer JTrd). For symmetry, the average relative error between attribution scores a_ij and a_ji is consistently below 24%, with clear diagonal symmetry observed. For stability, attribution scores remain stable across different core-set sizes, with relative errors below 30%.
> Initial observations suggest that optimizer settings and core-set size influence these properties.
>
> ### **Summary**
>  We acknowledge the limitations of our current work and will revise the manuscript to clarify assumptions, experimental comparisons, and better position our contributions. Thank you again for the valuable feedback.

---

### Official Review · Reviewer_JTrd · 2025-03-11

**Overall Recommendation:** 4

**Summary:**

This paper presents a novel coreset selection method, which minimizes the squared loss to balance contributions between coreset and non-coreset samples. The method follows a two step process: first select samples with the highest MRMC value, then train a proxy model to select samples to increase diversity. Extensive experiments on benchmarks demonstrate that the proposed method significantly accelerates training and reduces computational costs while improving model performance.

**Claims And Evidence:**

Yes

**Essential References Not Discussed:**

[1] Refined coreset selection: Towards minimal coreset size under model performance constraints. ICML 2024

[2] Mind the Boundary: Coreset Selection via Reconstructing the Decision Boundary. ICML 2024

**Experimental Designs Or Analyses:**

Yes

**Methods And Evaluation Criteria:**

Yes

**Other Comments Or Suggestions:**

1. The paper should empirically validate key assumptions made in the theoretical analysis. For instance, it is essential to verify whether the attribution values are indeed approximately symmetric as assumed in Section 4.3, and if they remain stable throughout training (as mentioned in line 207).
2. The Taylor expansion used in Eq. (6) assumes that the high order terms are negligible. Empirical study is needed to assess the impact of the residual terms on approximating $\Delta l_i^t$.

**Other Strengths And Weaknesses:**

Strengths:
1. The method is built on theoretical foundation, providing rigorous justification for the proposed approach.
2. The novel MRMC criterion is both technically sound and intuitive, which offers a novel perspective on evaluating sample importance.
3. The experimental analysis is thorough and demonstrates promising performance gains across multiple datasets.

Weaknesses:
1. Several works in coreset selection is missing, as mentioned in section above.
2. The assumption of symmetry in attribution values based on mini-batch SGD can be fragile, especially since it relies on a small learning rate, while training in the experiment begins with a high learning rate 0.1
3. GraNd score proposed in [1] should be included for comparison.

[1] Deep Learning on a Data Diet: Finding Important Examples Early in Training. NeurIPS 2021

**Questions For Authors:**

Please address the Comments section.
1. Following weakness 3, how does GraNd perform on the benchmarks?
2. In section 4.3, you argue that a small number of training epochs on full samples are sufficient. How does the MRMC's performance change when the number of training epochs $R$ exceeds 20?
3. Can you provide the architecture of the proxy model used for regularization and the hyperparameter for training the proxy model?

**Relation To Broader Scientific Literature:**

The paper introduces a novel coreset selection criterion that reframes sample importance via squared loss minimization. It extends prior gradient-based approaches by introducing the Maximum Reduction as Maximum Contribution (MRMC) metric to quantify each sample’s impact on training.

**Theoretical Claims:**

Yes

---

> ### Author Rebuttal · Authors · 2025-03-30
>
> Your suggestions are very helpful for improving the quality of this work. Below we address each of your concerns:
> ### **Response #1: Missing Related Work**
> Thank you for pointing this out. In the revised manuscript, we will add a discussion on “Refined Coreset Selection” (ICML 2024), which formulates the problem as a bi-level optimization and employs Lexicographic Optimization for solution. This work belongs to the class of optimization-based methods and provides an important reference.
> “Mind the Boundary” (ICML 2024) is already discussed in our original manuscript and also included as a baseline named "Boundary" in our main experiments (see Section 5.1 Competitors). We will clarify this more explicitly in the revised version.
> ### **Response #2: Empirical Validation for Assumptions**
>  We appreciate the reviewer’s observation. It was indeed an oversight on our part not to empirically validate the assumptions. To address this, we conducted a case study to assess the **symmetry and stability** of the attribution scores. Specifically, we randomly sampled 50 instances from CIFAR-100 and computed their attribution scores throughout the 10 training epochs under four different settings: using the full dataset and coresets of 30%, 50%, and 70%. This resulted in four 50*50 attribution matrices.  **Symmetry:** Visualizations of the attribution matrices consistently exhibit strong diagonal symmetry across all settings. The relative symmetry errors are 16%, 19%, 22%, and 24% for the 30%, 50%, 70% coresets, and full dataset, respectively—indicating that symmetry degrades as the amount of training data increases. **Stability:** We further normalized the attribution scores and computed the average relative error against the full-data attribution matrix. The relative errors for the 30%, 50%, and 70% core-sets are 27%, 26%, and 22%, respectively, suggesting improved stability with larger core-sets. (Due to the review policy, we are unable to include external links to visualizations.)
> ### **Response #3: Comparison with GraNd**
> GraNd, proposed in [1], is a well-known and efficient method, and EL2N is its approximation. As expected, the two methods yield similar experimental results. Following your suggestion, we implemented GraNd and will update the results in the revised manuscript. Similar to EL2N, GraNd’s performance drops significantly as the core-set size decreases. In contrast, our MRMC-R method consistently outperforms GraNd at 50% and 30% coreset sizes, especially on more challenging tasks. This demonstrates the strength of MRMC in scenarios where effective selection is needed.
> ### **Response #4: Proxy Model Details**
> The proxy model used in our method is the final linear classification layer (i.e., copy.deepcopy(model.classifier)). Analyzing gradients from the output layer is a common and efficient practice in coreset literature. As described in line 350, the proxy model is trained using SGD with a learning rate of 0.01, batch size of 512, for 10 epochs. Since it contains only one linear layer, training is very fast—even on large datasets like ImageNet-1K, it completes in under one minute.
> ### **Response #5: Training Epochs R  > 20**
> We originally evaluated R = 5, 10, and 20 in the sensitivity analysis, but did not include larger values. Thank you for pointing this out. To address this, we  conducted an experiment with R = 30. The results show that, except for the 70% and 50% core sets of CIFAR-100, performance generally degrades. This indicates that an excessively large R is inappropriate. This suggests that larger R may lead to overfitting in early training and reduce the effectiveness of the attribution signal.
>
> | Method    | C10(70%) | C10(50%) | C10(50%) | C100(70%) | C100(50%) | C100(30%) | TI(70%) | TI(50%) | TI(30%) |
> |:----------------:|:--------:|:--------:|:--------:|:---------:|:---------:|:---------:|:-------:|:-------:|:-------:|
> | MRMC-R (R=20)    | 95.46    | 95.24    | 93.13     | 77.17     | 74.46     | 68.66   | 60.42   | 57.03   | 50.88 |
> | MRMC-R (R=30)    | 95.35    | 95.16    | 93.00     | 77.83     | 74.99     | 66.38   | 60.25   | 56.32   | 49.66 |
> | GraNd            | 95.01    | 94.59    | 88.91     | 77.02     | 66.58     | 49.23   | 55.98   | 39.67   |  20.41 |
> | EL2N             | 95.40    | 95.23    | 91.94     | 77.61     | 67.60     | 32.55   | 56.08   | 38.96   |  13.79  |
>
>
> ### **Summary**
> We hope that the added analyses and new experimental comparisons comprehensively address your concerns.
>
> [1] Deep Learning on a Data Diet: Finding Important Examples Early in Training. NeurIPS 2021

---

> > ### Comment · Reviewer_JTrd · 2025-04-03
> >
> > Good work. My concerns are fully addressed.

---

### Official Review · Reviewer_hRSX · 2025-03-15

**Overall Recommendation:** 3

**Summary:**

The authors propose a new objective to select subsamples for training deep learning models. They call it MRMC (Maximum Reduction as Maximum Contribution) which essentially says that a data point that leads to more reduction in the squared error loss during the first few epochs of training relative to other points, is a more important point and should be a part of the subsamples used for further training. They also introduce a regularization term in their objective to take care of overfitting. They show the effectiveness of their sampling method compared to a variety of methods in their experiments on standard image datasets.

**Claims And Evidence:**

The paper is written clearly and is easy to follow. The claims are convincing enough.

**Essential References Not Discussed:**

Not to the best of my knowledge.

**Experimental Designs Or Analyses:**

Seems good but some more experiments may be added. See weaknesses.

**Methods And Evaluation Criteria:**

Yes.

**Other Comments Or Suggestions:**

See Weaknesses

**Other Strengths And Weaknesses:**

Strengths:
1) Subset selection is becoming a very important problem in the era of huge models and this paper would be of interest to a broad community.
2) The algorithm is intuitive and simple.
3) Experiments are presented on good number of datasets and show very good.

Weaknesses:
1)Although the theoretical intuition behind the idea and framework is clear, the method does not really give theoretical guarantees.
2) In the experiments section, the authors have not made comparisons with what they call the optimization-based methods like GradMatch. I think these comparisons are also required as they are essentially the methods with some theoretical guarantees. Even though I admire the detailed experiments authors have done, I believe this kind of study needs to be even more comprehensive.

**Questions For Authors:**

None. Please address the weaknesses

**Relation To Broader Scientific Literature:**

With the huge deep learning architectures being used in today's age, subsampling algorithms are very important. There are either algorithm with theoretical guarantees but practically tough to implement or some well performing heuristics without any guarantees. This paper tries to bridge the gap by providing a new subset selection objective which is theoretically motivated and grounded and at the same time good in practice.

**Theoretical Claims:**

There are not many theoretical claims in the pair. However, the theoretical intuition behind the main algorithm is clearly explained.

---

> ### Author Rebuttal · Authors · 2025-03-29
>
> We sincerely thank the reviewer for the thoughtful and fair assessment. We appreciate the opportunity to  address your concerns.
> ### **Response #1: Lack of Theoretical Guarantees**
> We acknowledge that our method does not provide formal theoretical guarantees. This work aims to bridge the gap between purely heuristic methods and theoretically grounded but computationally expensive approaches. Our method (closer to a heuristic method) is motivated by a clear theoretical intuition and achieves higher performance than other heuristic algorithms. We will provide more analysis and supporting empirical evidence to justify this intuition and our contributions in the revised manuscript.
> 1. **Complexity analysis** of MRMC and **runtime comparisons** (see Responses #1 and #4 to Reviewer *opAA*) to demonstrate its practical efficiency.
>
> 2. A **case study validating the assumption** that attribution scores are approximately symmetric and stable, which is fundamental to our method (see Response #2 to Reviewer *JTrd* ).
>
> While a full theoretical guarantee is absent, we believe this combination of intuition, analysis, and empirical validation provides a meaningful step forward in the design of practical and reliable core-set selection algorithms.
>
> ### **Response #2. Comparison with Optimization-based Methods**
> Thank you for highlighting the importance of comparison with optimization-based methods. We addressed this concern by attempting to reproduce GradMatch[1] and Glister[2] as optimization-based baselines. Unfortunately, we found that GradMatch (with GPU) was too computationally demanding—core-set selection time exceeded even full model training time in our setting. Other optimization-based methods also face this issue. Therefore, we selected Glister as a more practical baseline and adapted it by using only output-layer gradients to reduce its computational cost (selection time reduced to 27 seconds on CIFAR-100), but still much slower compared to MRMC-R (3s) and MRMC (0.2s).
>
> We have now added more comparisons (i.e.  Glister, GraNd) in the revised manuscript. The results show that MRMC achieves better core-set quality while being significantly more efficient. (C10 and C100 correspond to CIFAR-10 and CIFAR-100, respectively, and TI refers to Tiny-ImageNet. The numbers in parentheses indicate the core set size.)
>
>
> |         Method         | C10(70%) | C10 (50%) | C10 (30%) | C100 (70%) | C100 (50%) | C100(30%) | TI(70%) | TI(50%) | TI( 30%) |
> |:----------------------:|:-----------:|:-----------:|:-----------:|:------------:|:------------:|:------------:|:-----------------:|:-----------------:|:-----------------:|
> | MRMC                   |   94.94     |   94.61     |   92.27     |    77.76     |    74.43     |    67.28     |       59.89       |       56.89       |       49.99       |
> | MRMC-R                 |   95.46     |   95.24     |   93.13     |    77.17     |    74.46     |    68.66     |       60.42       |       57.03       |       50.88       |
> | Glister                |   94.80     |   94.20     |   90.15     |    76.80     |    73.21     |    66.90     |       57.20       |       55.05       |       48.23       |
>
> ### **Summary**
> We hope that the added analyses and new experimental comparisons comprehensively address your concerns and help position MRMC as an effective and scalable solution to the core-set selection problem. Thank you again for your thoughtful feedback.
>
> [1] GRAD-MATCH: Gradient Matching based Data Subset Selection for Efficient Deep Model Training,ICML,2021
>
> [2] GLISTER: Generalization based Data Subset Selection for Efficient and Robust Learning,AAAI,2021

---

### Decision · Program_Chairs · 2025-05-01

**Decision:**

Accept (poster)

**Comment:**

The reviewers appreciated the theoretically motivated and computationally efficient algorithm, and the solid experimental evaluation. Although the problem setting has been studied many times, this work is a solid contribution. Thus, I recommend acceptance.